# Melatonin Attenuates Oxidative Stress-Induced Apoptosis of Bovine Ovarian Granulosa Cells by Promoting Mitophagy via SIRT1/FoxO1 Signaling Pathway

**DOI:** 10.3390/ijms241612854

**Published:** 2023-08-16

**Authors:** Gaoqing Xu, Yangyunyi Dong, Zhe Wang, He Ding, Jun Wang, Jing Zhao, Hongyu Liu, Wenfa Lv

**Affiliations:** 1Jilin Provincial International Joint Research Center of Animal Breeding & Reproduction Technology, College of Animal Science and Technology, Jilin Agricultural University, Changchun 130118, China; 2Joint Laboratory of Modern Agricultural Technology International Cooperation, Ministry of Education, College of Animal Science and Technology, Jilin Agricultural University, Changchun 130118, China; 3Key Laboratory of Animal Production, Product Quality and Security, Ministry of Education, College of Animal Science and Technology, Jilin Agricultural University, Changchun 130118, China

**Keywords:** melatonin, oxidative stress, apoptosis, mitophagy, bovine ovarian granulosa cells

## Abstract

Oxidative-stress-induced apoptosis of granulosa cells is considered to be a main driver of follicular atresia. Increasing evidence suggests a protective effect of melatonin against oxidative damage but the mechanism remains unclear. The aim of this study is to investigate the effects of melatonin on mitophagy and apoptosis of bovine ovarian granulosa cells under oxidative stress, and to clarify the mechanism. Our results indicate that melatonin inhibited H_2_O_2_-induced apoptosis and mitochondrial injury of bovine ovarian granulosa cells, as revealed by decreased apoptosis rate, reactive oxygen species (ROS) levels, Ca^2+^ concentration, and cytochrome C release and increased mitochondrial membrane potential (ΔΨm). Simultaneously, melatonin promoted mitophagy of bovine ovarian granulosa cells through increasing the expression of PTEN-induced putative kinase 1 (PINK1), PARKIN, BECLIN1, and LC3II/LC3I; decreasing the expression of sequestosome 1 (SQSMT1); and promoting mitophagosome and lysosome fusion. After treatment with a mitophagy inhibitor CsA, we found that melatonin alleviated apoptosis and mitochondrial injury through promoting mitophagy in bovine ovarian granulosa cells. Furthermore, melatonin promoted the expression of silent information regulator 1 (SIRT1) and decreased the expression level of forkhead transcription factors class O (type1) (FoxO1). By treatment with an SIRT1 inhibitor EX527 or FoxO1 overexpression, the promotion of melatonin on mitophagy as well as the inhibition on mitochondrial injury and apoptosis were reversed in bovine ovarian granulosa cells. In conclusion, our results suggest that melatonin could promote mitophagy to attenuate oxidative-stress-induced apoptosis and mitochondrial injury of bovine ovarian granulosa cells via the SIRT1/FoxO1 signaling pathway.

## 1. Introduction

Bovine granulosa cells are important for the quality of oocytes and ovarian follicle development and ovulation, which seriously affect the reproductive performance of dairy cows and production efficiency of farms [1]. Most follicles are atretic, and less than 1% of the follicles can develop to the preovulation stage. Granulosa cells are the main functional cells in the follicle, which affect the growth, development, and maturation of the follicle in an autocrine and paracrine manner [2,3]. Granulosa cell apoptosis is the major driver of abnormal follicular atresia, and granulosa cell dysfunction causes premature ovarian failure and female subfertility [4]. Excessive accumulation of reactive oxygen species (ROS), as the main cause of oxidative stress, has been described as a key signaling molecule of apoptosis because the increase in mitochondrial permeability can cause the release of Ca^2+^ and Cytochrome C (Cyt-c), which can induce mitochondrial injury and apoptosis [5]. It has been found that oxidative stress can induce granulosa cell apoptosis, causing antral follicular atresia and affecting the quality of oocytes, which may explain the pathogenesis of anovulatory disorders [6]. Thus, clarifying the preventive mechanism of granulosa cell apoptosis would provide a reasonable treatment strategy for reproductive disorders caused by abnormal follicular atresia.

Melatonin (N-acetyl-5-methoxytryptamine) is a polytropic endogenous hormone of indoleamine, which has been found to be secreted in the ovary [7,8]. Additionally, melatonin receptors (MT1 and MT2) are expressed in the ovarian granulosa cells [9]. Melatonin participates in the regulation of ovarian function, including follicle formation, oocyte maturation, and reducing oxidative damage in granulosa cells [10]. Previous studies showed that melatonin could protect mice and porcine ovarian granulosa cells against apoptosis and oxidative stress [11,12]. Simultaneously, melatonin can maintain mitochondrial function based on its antioxidant, anti-apoptotic, and free radical scavenging activities [13,14]. Nevertheless, the molecular mechanism of melatonin regulating mitochondrial injury and apoptosis is unclear.

Mitophagy, a process of selectively removing redundant or damaged mitochondria, is necessary to maintain mitochondrial function and cellular health [15]. External stimuli such as ROS, nutrient deficiency, and cell aging lead to the decline of mitochondrial membrane potential (ΔΨm) and depolarization damage, thereby inducing mitophagy. In early damage, oxidative stress results in the accumulation of PTEN-induced putative kinase 1 (PINK1) on damaged mitochondria; then, it leads to recruitment of the PARKIN protein into damaged mitochondria. Subsequently, mitochondria are encapsulated by autophagosomes into mitophagosomes, which fuse with lysosomes into mitophagolysosomes to finish mitochondria degradation [16,17,18]. However, severe mitochondrial damage activates apoptosis proteinase to prevent mitophagy and activate apoptosis, resulting in cell death [19]. Studies have reported that mitophagy plays an important role in mitochondrial injury related diseases, such as cardiac ischemia reperfusion (I/R) and Alzheimer’s disease [15,20]. Importantly, melatonin was found to promote PINK1/PARKIN-mediated mitophagy [20]. Nevertheless, the role of mitophagy in the protection of melatonin from H_2_O_2_-induced apoptosis of bovine ovarian granulosa cells is unknown.

Silent information regulator 1 (SIRT1), a nicotinamide adenine dinucleotide (NAD+)-dependent histone deacetylase, has an indispensable role in several physiological activities, including oxidative stress, aging, apoptosis, and mitochondrial damage. SIRT1 can be regulated directly by melatonin, and melatonin has been reported to inhibit apoptosis and oxidative stress by a SIRT1-dependent mechanism [21]. Moreover, SIRT1 can decrease the activity of forkhead box protein O1 (FoxO1) through deacetylating FoxO1 to play a crucial role in oxidative damage [22,23]. Studies have reported that FoxO1 is involved in the growth and development, metabolism, and tumorigenesis of organisms by regulating various physiological processes including oxidative stress, proliferation, and apoptosis [24,25,26]. In mammals, FoxO1 induces granulosa cell apoptosis and damages follicular development by enhancing the transcriptional activity of apoptotic factors and has a high expression level in the granulosa cells of atretic follicles [27]. Interestingly, FoxO1 plays a vital role in the regulation of mitophagy [28]. Moreover, melatonin was found to alleviate oxidative damage of mouse granulosa cells through FoxO1-mediated autophagy via the SIRT1-FoxO1-ATG7 pathway [29]. However, whether SIRT1/FoxO1 pathway is involved in melatonin regulating mitophagy needs further study.

Here, we determined the effects of melatonin on apoptosis, mitochondrial injury and mitophagy in bovine ovarian granulosa cells. Importantly, we investigated the role of mitophagy in the protection of melatonin on bovine ovarian granulosa cells and the mechanism of melatonin regulating mitophagy. Our study may provide a new viewpoint into the protective mechanism of melatonin and the treatment strategy for reproductive disorders.

## 2. Results

### 2.1. Melatonin Suppresses H_2_O_2_-Induced Apoptosis of Bovine Ovarian Granulosa Cells

We treated bovine ovarian granulosa cells with melatonin (0, 0.01, 0.1, 1, or 10 μM) for 24 h or H_2_O_2_ (0, 50, 100, 200, 300, 400, 500, or 1000 μM) for 4 h and examined cell viability. As depicted in Figure 1A,B, melatonin at the tested concentrations was generally safe (*p* > 0.05) and H_2_O_2_ decreased cell viability in a dose-dependent manner for the granulosa cells (*p* < 0.001). It was observed that 400 μM H_2_O_2_ caused approximately 50% loss of cell viability. Thus, we chose 400 μM H_2_O_2_ for follow-up treatment. Next, we treated cells with melatonin (0, 0.01, 0.1, 1, or 10 μM) for 24 h and then added 400 μM H_2_O_2_ for 4 h. As depicted in Figure 1C, compared with the H_2_O_2_-treated group, melatonin increased granulosa cell viability and 0.01 μM melatonin was used for follow-up experiments (*p* < 0.05).

Subsequently, we further examined the effect of melatonin on apoptosis. The results of qRT-PCR analysis showed a marked increase in the mRNA levels of proapoptotic factors BAX and CASPASE-3 in the H_2_O_2_-treated group, while the anti-apoptotic factor BCL-2 was decreased (Figure 1D). Similarly, the protein levels of BAX and CASPASE-3 were suppressed and BCL-2 level were promoted by melatonin pretreatment (Figure 1E). Additionally, we detected apoptosis of granulosa cells by flow cytometry. Our results showed that H_2_O_2_ treatment increased the apoptosis rate of granulosa cells while melatonin decreased the apoptosis rate (*p* < 0.05, Figure 1G,H). TUNEL assay also indicated that compared with the H_2_O_2_-treated group, melatonin significantly reduced the percentage of TUNEL-positive cells (Figure 1I,J). The above results suggest that melatonin suppressed H_2_O_2_-induced apoptosis of bovine ovarian granulosa cells.

### 2.2. Melatonin Ameliorates H_2_O_2_-Induced Mitochondrial Injury and Mitophagy Deficits of Bovine Ovarian Granulosa Cells

We examined the mean fluorescence intensity by flow cytometry to detect ROS levels. As depicted in Figure 2A, the mean fluorescence intensity of H_2_O_2_-treated granulosa cells was significantly increased, while melatonin pretreatment decreased the mean fluorescence intensity (*p* < 0.001). Next, we examined ΔΨm by flow cytometry; the results showed that H_2_O_2_ treatment decreased the ΔΨm of bovine ovarian granulosa cells and melatonin attenuated H_2_O_2_-induced ΔΨm depolarization (Figure 2B,C). We also detected the concentrations of intracellular Ca^2+^ and released cytochrome C to assess the protection of melatonin against mitochondrial dysfunction. As depicted in Figure 2D,E, melatonin alleviated the promotion of Ca^2+^ influx and cytochrome C release by H_2_O_2_ (*p* < 0.05). Interestingly, H_2_O_2_ decreased the relative protein levels of PINK1, PARKIN, BECLIN1, and LC3II/LC3I and increased the relative level of SQSMT1, while melatonin pretreatment reversed these relative protein levels (Figure 2G). Similarly, H_2_O_2_ treatment inhibited mitophagosome and lysosome fusion of bovine ovarian granulosa cells, while melatonin significantly attenuated the inhibition of mitophagosome and lysosome fusion by H_2_O_2_ (Figure 2H,I). Taken together, melatonin ameliorated mitochondrial injury and mitophagy deficits of bovine ovarian granulosa cells.

### 2.3. Mitophagy Promotion by Melatonin Attenuates H_2_O_2_-Induced Mitochondrial Injury and Apoptosis in Bovine Ovarian Granulosa Cells

To verify the role of mitophagy in melatonin alleviating H_2_O_2_-induced mitochondrial injury and apoptosis, we used CsA to treat bovine ovarian granulosa cells. We examined mitophagy-related expression, and mitophagosome and lysosome fusion. As depicted in Figure 3A–D, compared with the MT+H_2_O_2_ group, CsA pretreatment decreased the relative protein levels of PINK1, PARKIN, BECLIN1, and LC3II/LC3I; increased SQSMT1 level; and inhibited mitophagosome and lysosome fusion (*p* < 0.01). These results suggested that CsA pretreatment could block melatonin-promoted mitophagy of bovine ovarian granulosa cells. Next, we detected mitochondrial function after CsA pretreatment. We found that, compared with the MT+H_2_O_2_ group, CsA pretreatment increased ROS level (Figure 3E), Ca^2+^ influx (Figure 3H), and cytochrome C release (Figure 3I) and decreased the ΔΨm of bovine ovarian granulosa cells (Figure 3F,G). Moreover, as depicted in Figure 3J–L, CsA pretreatment increased the mRNA and protein levels of BAX and CASPASE-3, and decreased the level of BCL-2 (*p* < 0.05). Simultaneously, CsA significantly increased the apoptosis rate (Figure 3M,N) and the ratio of TUNEL-positive cells (Figure 3O,P). Our findings suggest that melatonin attenuated H_2_O_2_-induced mitochondrial injury and apoptosis by promoting mitophagy in bovine ovarian granulosa cells.

### 2.4. SIRT1 Activation by Melatonin Promotes Mitophagy to Attenuate H_2_O_2_-Induced Mitochondrial injury and Apoptosis in Bovine Ovarian Granulosa Cells

To reveal the molecular mechanisms of melatonin promoting mitophagy of bovine ovarian granulosa cells, we demonstrated whether melatonin activated SIRT1 expression. The results showed that the protein level of SIRT1 in the MT+H_2_O_2_ group was higher than that in the H_2_O_2_ group (Figure 4A,B). To investigate the role of SIRT1 in the promotion of mitophagy by melatonin, we verified an inhibitor of SIRT1 by pretreatment with EX527. The staining results showed that EX527 pretreatment inhibited mitophagosome and lysosome fusion (Figure 4C,D). Simultaneously, the results of Western blot indicated that EX527 pretreatment decreased the relative levels of PINK1, PARKIN, BECLIN1, and LC3II/LC3I and increased SQSMT1 level (Figure 4E,F), which indicated that melatonin ameliorated H_2_O_2_-induced mitophagy deficits by activating SIRT1. Subsequently, we detected the effects of SIRT1 inhibition on mitochondrial injury and apoptosis. As depicted in Figure 5A–E, EX527 pretreatment reversed the protection of melatonin against mitochondrial injury, manifested by increasing ROS levels, promoting Ca^2+^ influx and Cyt-c release, and decreasing ΔΨm of bovine ovarian granulosa cells (*p* < 0.05). Moreover, as depicted in Figure 5F–H, EX527 pretreatment significantly increased the expression of BAX and CASPASE-3, and decreased BCL-2 expression (*p* < 0.05). Simultaneously, EX527 increased the apoptosis rate (Figure 5I,J) and ratio of TUNEL-positive cells (Figure 5K,L). The above results suggest that melatonin activated SIRT1 to promote mitophagy, thereby attenuating H_2_O_2_-induced mitochondrial injury and apoptosis in bovine ovarian granulosa cells.

### 2.5. SIRT1/FoxO1 Activation by Melatonin Promotes Mitophagy to Attenuate H_2_O_2_-Induced Mitochondrial injury and Apoptosis in Bovine Ovarian Granulosa Cells

Furthermore, considering that SIRT1 activation can deacetylate FoxO1, we examined the role of the SIRT1/FoxO1 signaling pathway in the promotion of melatonin on mitophagy. We found that H_2_O_2_ treatment increased FoxO1 protein level while melatonin treatment alleviated FoxO1 expression level (Figure 6A,B). After EX527 pretreatment, FoxO1 expression significantly upregulated compared with the MT+H_2_O_2_ group (Figure 6C,D). Subsequently, we performed subsequent experiments by overexpressing FoxO1. As depicted in Figure 6E,F, after FoxO1 overexpression, the FoxO1 expression level was higher than that in the MT+H_2_O_2_ group (*p* < 0.01). Importantly, compared with the NC+MT+H_2_O_2_ group, FoxO1 overexpression downregulated the expression of PINK1, PARKIN, BECLIN1, and LC3II/LC3I and upregulated SQSMT1 expression (*p* < 0.05). Simultaneously, FoxO1 overexpression inhibited mitophagosome and lysosome fusion (Figure 6I,J).

Additionally, we examined the effect of FoxO1 overexpression on apoptosis of bovine ovarian granulosa cells. Our results showed that FoxO1 overexpression reversed the protection of melatonin against mitochondrial injury, manifested by increased ROS levels (Figure 7A), Ca^2+^ influx (Figure 7D), and Cyt-c release (Figure 7E) and decreased ΔΨm (Figure 7B,C) of bovine ovarian granulosa cells (*p <* 0.05). Moreover, compared to the NC+MT+H_2_O_2_ group, FoxO1 overexpression upregulated the expression of BAX and CASPASE-3, and downregulated BCL-2 expression (Figure 7F–H). Similarly, after FoxO1 overexpression, the apoptosis rate (Figure 7I,J) and the ratio of TUNEL-positive cells (Figure 7K,L) significantly increased compared to the NC+MT+H_2_O_2_ group (*p <* 0.05). Above all, our results suggested that the SIRT1/FoxO1 pathway might be responsible for melatonin’s promotion of mitophagy to attenuate H_2_O_2_-induced mitochondrial injury and apoptosis in bovine ovarian granulosa cells.

## 3. Discussion

To elucidate the mechanism of melatonin anti-oxidation and anti-apoptosis is of great significance for understanding follicular atresia. In this study, melatonin attenuates oxidative-stress-induced mitochondrial injury and apoptosis, and promotes PINK1/PARKIN-mediated mitophagy of bovine ovarian granulosa cells. Importantly, our research provides evidence that melatonin inhibits oxidative-stress-induced apoptosis of bovine ovarian granulosa cells by promoting mitophagy via the SIRT1/FoxO1 signaling pathway.

It is well known that melatonin is considered a potential antioxidant. Melatonin could decrease ROS levels and malondialdehyde (MDA), and increase superoxide dismutase (SOD) and glutathione peroxidase (GSH-Px), thereby inhibiting oxidative stress [21]. Importantly, it has been reported that melatonin can inhibit cell apoptosis [30]. In this study, melatonin attenuated H_2_O_2_-induced apoptosis of bovine ovarian granulosa cells. A study supporting our results showed that melatonin reduced the apoptosis level in rat peripheral blood lymphocytes, accompanied by the decreased BAX expression and increased BCL-2 expression [31]. In endometrial stromal cells, melatonin can inhibit apoptosis by downregulating the ratio of TUNEL-positive cells [32]. In the fetal heart, melatonin downregulates the expression of BAX and CASPASE-3 to inhibit oxidative-stress-induced apoptosis [33]. Melatonin could reverse oxidant stress and apoptosis of mouse oocytes in polycystic ovary syndrome (PCOS) [34]. Moreover, melatonin could attenuate postovulatory oocyte dysfunction to delay oocyte aging by regulating SIRT1 expression [35]. It has also been reported that melatonin could regulate granulosa cell proliferation and steroidogenesis to modulate swine ovarian follicle function; further, melatonin could increase antioxidant power and inhibit steroidogenesis of luteal cells to modulate swine luteal functions, which suggests that melatonin could regulate ovary function by acting on granulosa cells [36,37]. In our results, although H_2_O_2_ increased the apoptosis rate, which was decreased by melatonin, the apoptosis rate was relatively low. The low apoptosis rate may be mainly due to the limitations of the in vitro culture mode, and the state of adherent cells cultured in vitro usually differs from that of primary cells, thus requiring a large number of in vivo experiments in the future.

Additionally, ROS play an important role in mitochondrial injury and apoptosis [38,39]. The excessive production of ROS can decrease ΔΨm to cause the release of pro-apoptotic factors including caspase, Ca^2+^, and Cyt-c, thus inducing apoptosis [5,40]. Melatonin can scavenge free radicals (e.g., reduce ROS levels) to prevent cellular mitochondrial injury, thereby alleviating oxidative damage [41,42]. Melatonin could also improve ΔΨm and antioxidant enzyme levels, and decrease ROS levels, Ca^2+^ concentration, and cytochrome C release to suppress H_2_O_2_-induced apoptosis of rooster Leydig cells [43]. Consistently, it has been demonstrated that melatonin can repress mitochondria dysfunction and apoptosis in Sudan-I-exposed mouse oocytes by inhibiting ROS production and increasing ΔΨm [34]. In our study, melatonin alleviated mitochondrial injury by decreasing ROS levels, improving ΔΨm, and reducing the levels of Ca^2+^ and Cyt-c, which suggested that melatonin might inhibit oxidative-stress-induced apoptosis of bovine ovarian granulosa cells via a mitochondrial-dependent mechanism.

Mitophagy is an essential physiological activity for maintaining cell health by clearing away damaged and redundant mitochondria [44]. However, when mitochondria are severely damaged, the activation of apoptotic proteinase can prevent mitophagy and cause apoptosis [19]. High-glucose (50 mM) or H_2_O_2_ treatments were found to decrease PINK1 and PARKIN expression, and high glucose could inhibit cell proliferation and promote apoptosis by causing mitophagy defects in retinal pigment epithelium [45]. Studies also reported that cardiac ischemia reperfusion (I/R) injury could cause severe oxidative damage and inhibit mitophagy, as revealed by suppressed mitochondrial fusion and expression of LC3II and BECLIN1 [20,46]. Similarly, mitophagy was also found to be impaired in a rat model of spina bifida aperta, accompanied by downregulating the expression of LC3II/LC3I and PINK1 [47]. In our results, H_2_O_2_-induced oxidative damage disrupted mitophagosome and lysosome fusion; decreased the expression of PINK1, PARKIN, BECLIN1, and LC3II/LC3I; and increased the expression of SQSMT1, thus inhibiting mitophagy of bovine ovarian granulosa cells.

Importantly, increasing evidence indicates that melatonin promotes mitophagy to relieve cell damage. Melatonin has been found to alleviate radiculopathy-induced apoptosis and inflammation by promoting mitophagy, and si-PARKIN or CsA treatments reverse the protective effects of melatonin [48]. Melatonin could upregulate mitophagy, and reduce mitochondrial DNA damage and mitochondrial dysfunction by increasing PINK1 and PARKIN expression, thus alleviating liver fibrosis in rats [49]. In Alzheimer’s disease, melatonin was found to ameliorate mitophagy defects and mitochondrial dysfunction of mouse hippocampus by promoting mitophagosome and lysosome fusion [50]. Consistently, melatonin improves mitochondrial fusion/mitophagy to attenuate myocardial I/R injury [20]. Melatonin also inhibits ROS production, upregulates the expression of PARKIN, and maintains mitophagy to inhibit mitochondrial apoptosis, thereby promoting peripheral nerve repair [51]. All the above support our result that melatonin promotes PINK1/PARKIN-mediated mitophagy to alleviate oxidative-stress-induced apoptosis and mitochondrial injury in bovine ovarian granulosa cells.

Conversely, some studies reported that melatonin could inhibit mitophagy to improve oxidative damage [52,53]. Of course, the dual effects of mitochondrial damage on the promotion or inhibition of mitophagy and the dual regulation of melatonin on mitophagy have also been reported, respectively [54,55]. However, the specific mechanism of the dual effects of mitophagy requires further study.

Moreover, it is generally accepted that SIRT1 participates in the protection of melatonin, and melatonin has been described as an activator of SIRT1 [56]. In the present study, SIRT1 was downregulated under oxidative stress, while melatonin restored SIRT1 expression. Our results also demonstrated that melatonin reduced ROS production, and inhibited apoptosis and mitochondrial injury, of bovine ovarian granulosa cells in a SIRT1-dependent manner. A prior study reported that melatonin activated SIRT1 to inhibit apoptosis and oxidative stress in TM3, while treatment with the SIRT1 inhibitor EX527 reversed the protective effects of melatonin [21]. Additionally, melatonin activated SIRT1 to alleviate mitochondrial membrane damage and maintain mitochondrial homeostasis, thus inhibiting apoptosis in human granulosa cells [57]. Similarly, melatonin reduced ROS production and improved mitochondrial injury to abrogate rotenone-induced mitochondrial deficiency by enhancing SIRT1 expression, inhibiting oxidative stress and apoptosis in early porcine embryos, while SIRT1 knockdown abolished melatonin’s protection [58]. Importantly, SIRT1 can regulate mitophagy to renew defective mitochondria, maintaining mitochondrial health [59]. It was reported that SIRT1 activation upregulated the related markers of autophagy and PINK1/PARKIN-dependent mitophagy against oxidative stress in porcine intestinal epithelial cells, and EX527 treatment reversed these effects [60]. Several articles reported that SIRT1 could take part in the regulation of mitophagy by melatonin [52,61]. In diabetic rats or Leydig cells under high glucose, melatonin improved mitophagy defects to inhibit mitochondrial injury and oxidative stress by activating protein kinase (AMPK)/SIRT1 activity [62]. Above all, these results supported our findings that melatonin promotes PINK1/PARKIN-mediated mitophagy to inhibit H_2_O_2_-induced apoptosis and mitochondrial injury of bovine ovarian granulosa cells via a SIRT1-dependent mechanism.

Furthermore, SIRT1 can inhibit oxidative stress and apoptosis through deacetylation modification on FoxO1 [63]. Notably, FoxO1 has been described as an essential downstream target of melatonin in regulating oxidative stress and autophagy in mouse granulosa cells [29]. A study found that melatonin could relieve oxidative stress of mouse testes through the SIRT1/FoxO1 signaling pathway [64]. In aging mice, melatonin could promote hippocampal homeostasis by activating the SIRT1/FoxO1 signaling pathway [65]. Additionally, several studies found that SIRT1/FoxO1 signaling could be activated by physical exercise to improve PINK1/PARKIN-mediated mitophagy in Alzheimer’s disease [66,67,68]. In the present study, melatonin downregulated FoxO1 expression, while EX527 reversed the effect. We also found that FoxO1 overexpression abolished the protection of melatonin on bovine ovarian granulosa cells, suggesting that melatonin could promote mitophagy to inhibit apoptosis via the SIRT1/FoxO1 signaling pathway. However, a large number of in vivo experiments is required to verify the regulation of melatonin on female reproduction.

## 4. Materials and Methods

### 4.1. Cells Isolation and Culture

The fresh bovine ovaries obtained from the slaughterhouses were put into a thermos containing 37 °C saline and transported back to the operation room within 4 h. First, we removed excess tissue around the ovaries and washed them 5 times with 37 °C saline. Next, these ovaries were placed in a 37 °C water bath; then, we used a 5 mL syringe to aspirate the follicle fluid from the ovarian follicles with a diameter of 3–6 mm to obtain sufficient bovine ovarian granulosa cells. Cells were collected and washed with DMEM/F12 (Gibco, New York, NY, USA). A hemocytometer was used for cell counting. Cells were cultured in DMEM/F-12 with 10% fetal bovine serum (FBS) and 1% penicillin/streptomycin (Sangon Biotech, Shanghai, China); then, they were transferred to a 60 mm petri dish at a density of 1–1.2 × 10^6^ cells. When the primary cells adhere to the wall by more than 80%, cells were passaged to F1 generation and then treated based on the experimental design.

### 4.2. Cell Treatments

To screen the effective concentrations, bovine ovarian granulosa cells were treated with H_2_O_2_ (0, 50, 100, 200, 300, 400, 500, or 1000 μM, Sigma-Aldrich, Burlington, MA, USA) for 4 h or melatonin (0, 0.01, 0.1, 1, or 10 μM, Sigma-Aldrich, MA, USA) for 24 h [43,69]. Then, ovarian granulosa cells were treated with the optimal melatonin for 24 h before H_2_O_2_ exposure. Additionally, ovarian granulosa cells were treated with 1 μM Cyclosporin A (CsA, mitophagy inhibitor, ApexBio Technology, Houston, TX, USA) for 24 h or 10 μM Selisistat (EX527, SIRT1 inhibitor, MedChemExpress, NJ, USA) for 2 h before being treated with melatonin and H_2_O_2_. In overexpression treatment, the Flag-tagged FoxO1 plasmids or an empty control plasmid were individually transfected into bovine ovarian granulosa cells for 48 h.

### 4.3. Cell Viability Assay

Cell viability in different groups was analyzed by performing CCK-8 assay using a kit (ApexBio Technology, TX, USA). After 8 × 10^3^ cells/well (100 μL) were inoculated in 96-well plates and treated following the protocol, 10 μL of CCK-8 assay solution was added to each well. Subsequently, the 96-well plates were placed back in the incubator and incubated for 4 h, and all bubbles in each well were removed by slight shaking before detection. Next, the OD value was detected at 450 nm wavelength in a microplate reader (BioTek, Winooski, VT, USA).

### 4.4. RNA Extraction and qRT-PCR Analysis

Total RNA was obtained from bovine ovarian granulosa cells using Trizol reagent (Takara, Tokyo, Japan) following the instructions, 1 μg of which was reverse transcribed into cDNA using a PrimeScript RT Reagent Kit (Takara, Tokyo, Japan). Next, qRT-PCR was executed by a SYBR^®^Premix Ex TaqTM II (Takara, Tokyo, Japan) on Agilent StrataGene Mx3005P Detection System (Santa Clara, CA, USA). The primer information of BAX, CASPASE-3, BCL-2, and β-actin is shown in Table 1. The data were analyzed using the 2^−ΔΔCT^ method, and β-actin was used as the reference gene. This experiment was performed three times independently.

### 4.5. Western Blot Analysis

After treatment following the protocol, cells were gathered and lysed thoroughly in RIPA lysis buffer (Beyotime, Shanghai, China) for 20 min and then centrifuged at 12,000× *g* for 10 min to obtain the supernatant. The extracted protein concentration was measured using the enhanced BCA protein assay kit (Beyotime, Shanghai, China). A total 20 μg of protein from different groups was injected into each well for SDS-PAGE and transferred to nitrocellulose (NC) membranes (Millipore, Burlington, MA, USA) by semidry blotting. Subsequently, NC membranes were successively incubated in blocking buffer (LI-COR, Lincoln, NE, USA) for 1.5 h, followed by primary antibodies overnight and secondary antibodies for 1 h. The relative information of antibodies is shown in Table 2. The bands were visualized with enhanced chemiluminescence (ECL) reagent (NCM Biotech, Suzhou, China). Next, the bands were detected by a chemiscope image system (CLiNX Science instruments, Shanghai, China) and β-actin was used to normalize the density values of the bands. The value for the control group was set as 100%.

### 4.6. Apoptosis Analysis by Flow Cytometry

Apoptosis rate was evaluated with an Apoptosis Detection Kit (BD Biosciences, San Jose, CA, USA). After treatment following the protocol, bovine ovarian granulosa cells were gathered; suspended in 1× buffer with Annexin V-FITC and propidium iodide dyes with constant shaking to suspend the cells so as to ensure adequate binding of the cells to dyes; and measured by flow cytometry (ACEA Biosciences, Hangzhou, China). About 30,000 cells were analyzed in the cytometer. The flow cytometry results divide the cells into four quadrants, including Q2-1 quadrant (death cells), Q2-2 quadrant (late apoptotic cells), Q2-3 quadrant (normal cells), and Q2-4 quadrant (early apoptotic cells) [70]. In this experiment, the sum of the proportion of Q2-2 and Q2-4 was calculated as the apoptosis rate.

### 4.7. TUNEL Assay

The apoptosis of bovine ovarian granulosa cells was examined via a TUNEL assay. First, ovarian granulosa cells were seeded on cell climbing slices in 24-well plates. After ovarian granulosa cells were treated according to the protocol, cells were incubated in 4% paraformaldehyde (Millipore, USA) followed by 1% Triton X-100 (Sigma Aldrich, St. Louis, MO, USA) for 30 min. Next, TUNEL staining of fixed cells was performed with a TUNEL kit (Beyotime, China). Finally, the cell climbing slices were taken out of the 24-well plate and placed on a glass slide for observation by a cell imaging multifunctional detection system (Biotek Cytation5, Shoreline, WA, USA). The ratio of TUNEL-positive cells was calculated based on double staining of TUNEL (green) and DAPI (blue).

### 4.8. Mitophagy Assay

Ovarian granulosa cells were cultured on cell climbing slices in a 24-well plate. Mitophagy were detected with a Detection Kit (Dojindo Molecular Technologies, Rockville, MD, USA). First, cells were incubated in 100 nmol/L Mtphagy Dye working solution for 30 min. Next, cells were incubated in 1 μmol/L Lyso Dye working solution. Then, the cell climbing slices were taken out from the 24-well plate and placed on a glass slide. Finally, mitophagy was evaluated by a cell imaging multifunctional detection system.

### 4.9. ROS Assay

The ROS level was measured using a ROS Assay Kit (Beyotime, China) via flow cytometry. After being collected and washed, ovarian granulosa cells were suspended in DCFH-DA solution for 20 min following the manufacturer’s instructions. Next, granulosa cells were washed to sufficiently remove the remaining DCFH-DA. Then, about 30,000 cells from each group were detected using flow cytometry with the FITC channel at 488 nm, and mean fluorescence intensity was recorded to represent the ROS level.

### 4.10. ΔΨm Assay

ΔΨm was measured with an Assay Kit (BD Biosciences, USA) via flow cytometry. According to standard procedures, ovarian granulosa cells were collected and treated with 10 μg/mL of JC-1 for 20 min and protected from light. After being washed twice, granulosa cells were detected by flow cytometry. When ΔΨm is high, JC-1 aggregates to form polymers and emit red fluorescence (590 nm). Conversely, ΔΨm collapse caused JC-1 to diffuse, thereby forming monomers and emitting green fluorescence (527 nm). The percentage of mitophagy-positive cells was calculated based on double staining of lyso dye (green) and mtphage dye (red).

### 4.11. Intracellular Ca^2+^ Concentration Assay

Intracellular Ca^2+^ concentration was measured with the fluo-4 acetoxymethyl ester (Fluo-4 AM) Ca^2+^ probe (Beyotime, China) via flow cytometry. After treatment following the protocol, bovine ovarian granulosa cells were gathered and exposed to 250 μL of Fluo-4 AM for 30 min and then washed twice. Next, about 30,000 cells from each group were measured using flow cytometry with the FITC channel at 488 nm.

### 4.12. Cyt-c Release Assay

The concentration of Cyt-c was detected via ELISA with a Cyt-c ELISA Detection Kit (Mlbio, Shanghai, China). After treatment following the protocol, cell supernatant was centrifuged at 1000× *g* for 20 min. According to the manufacturer’s procedures, 50 μL of supernatant and recognition antigen were transferred to the antibody-coated wells successively. After incubating for 1 h and washing three times, avidin-HRP was added to incubate all wells for 30 min. Next, all wells were treated with chromogen solution and stop solution, and the OD value of them was detected at 450 nm via a microplate reader.

### 4.13. Statistical Analysis

The data were shown as mean ± standard deviation (SD). Differences between groups were determined by Student’s *t*-test and one-way analysis of variance (ANOVA). A difference with *p* < 0.05 was confirmed to be significant. All statistical data were analyzed by GraphPad Prism version 5.0 (San Diego, CA, USA).

## 5. Conclusions

To sum up, we found that melatonin inhibited apoptosis of bovine ovarian granulosa cells through improving mitochondrial injury. Notably, this may be the first report to reveal the activation of mitophagy by melatonin on bovine ovarian granulosa cells to inhibit apoptosis and to clarify the potential mechanism. Our results suggested that melatonin could attenuate oxidative-stress-induced apoptosis and mitochondrial injury of bovine ovarian granulosa cells by promoting PINK1/PARKIN-mediated mitophagy via the SIRT1/FoxO1 signaling pathway.

## Figures and Tables

**Figure 1 ijms-24-12854-f001:**
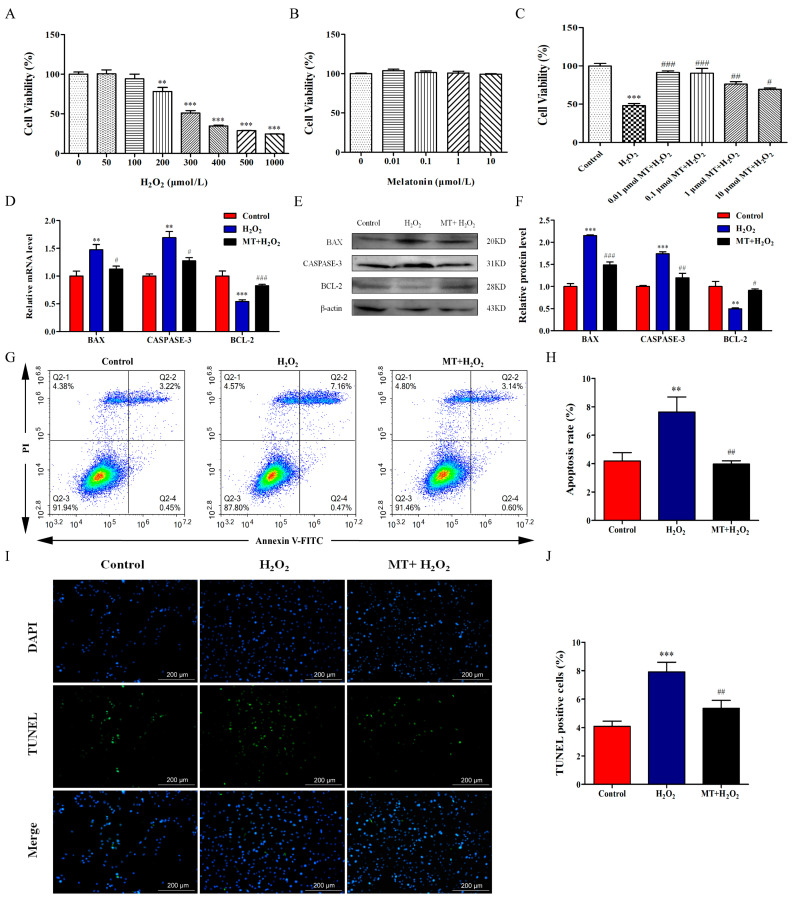
Melatonin represses H_2_O_2_-induced apoptosis of bovine ovarian granulosa cells. (**A**) Cell viability was determined using a CCK-8 kit after cells were treated with H_2_O_2_ (0, 50, 100, 200, 300, 400, 500, 1000 μM) (*n* = 6). (**B**) Cell viability was determined after cells were treated with melatonin (0, 0.01, 0.1, 1, 10 μM) (*n* = 6). (**C**) After granulosa cells were cultured with 0.01 μM melatonin for 24 h followed by 400 μM H_2_O_2_ for 2 h, cell viability was measured by a CCK-8 kit (*n* = 6). (**D**) The relative mRNA levels of BAX, CASPASE-3, and BCL-2 were detected by qRT-PCR (*n* = 3). (**E**,**F**) The relative protein levels of BAX, CASPASE-3, and BCL-2 were detected by Western blot (*n* = 3). β-actin was used to normalize the density values of the bands. (**G**,**H**) The apoptosis level of granulosa cells was measured by flow cytometry assay (*n* = 3). Apoptosis rate was expressed as the sum of the values of Q2-2 (late apoptotic cells) and Q2-4 (early apoptotic cells) in the four quadrants. The color from red to blue indicates a decrease in cell density. (**I**,**J**) The ratio of TUNEL-positive cells was detected by double staining of TUNEL (green) and DAPI (blue) (*n* = 3). The values are shown as mean ± SD. Differences between groups were determined by ANOVA. ** *p* < 0.01, or *** *p* < 0.001 vs. Control group (0 μmol/L melatonin group). # *p* < 0.05, ## *p* < 0.01, or ### *p* < 0.001 vs. H_2_O_2_ group.

**Figure 2 ijms-24-12854-f002:**
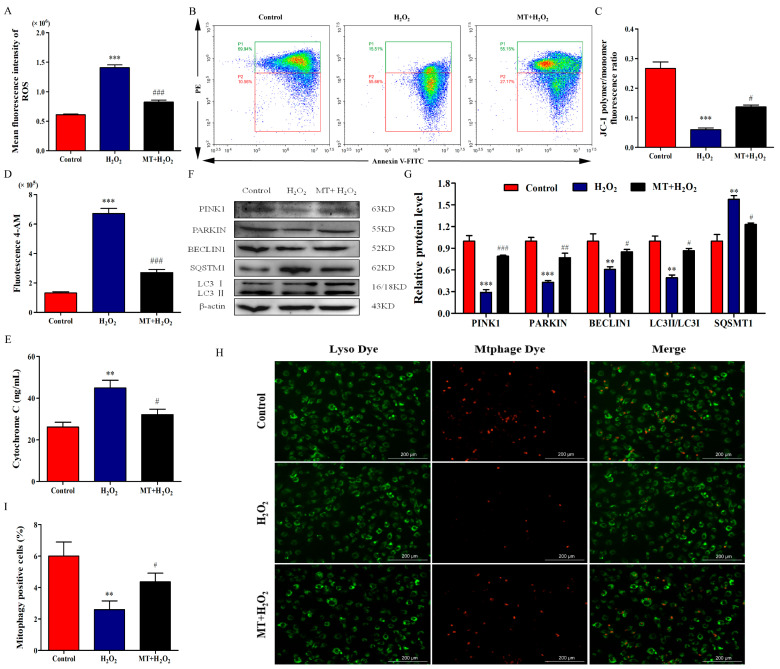
Melatonin ameliorates H_2_O_2_-induced mitochondrial injury and mitophagy deficit of bovine ovarian granulosa cells. (**A**) Mean fluorescence intensity was measured to detect ROS level by flow cytometry assay (*n* = 3). (**B**,**C**) ΔΨm was detected using the dye JC-1 (*n* = 3). The conversion of polymers and monomers, and the fluorescence ratio, were measured to analyze ΔΨm (*n* = 3). (**D**) The concentrations of intracellular Ca^2+^ were detected by flow cytometry assay using Fluo-4 AM (*n* = 3). (**E**) The release of Cytochrome C was measured by ELISA (*n* = 3). (**F**,**G**) The relative protein levels of PINK1, PARKIN, BECLIN1, SQSMT1, LC3I, and LC3II were detected by Western blot (*n* = 3). β-actin was used to normalize the density values of the bands. (**H**,**I**) The percentage of mitophagy-positive cells was detected by double staining of lyso dye (green) and mtphage dye (red). The values are shown as mean ± SD. Differences between groups were determined by ANOVA. ** *p* < 0.01, or *** *p* < 0.001 vs. Control group. # *p* < 0.05, ## *p* < 0.01, or ### *p* < 0.001 vs. H_2_O_2_ group.

**Figure 3 ijms-24-12854-f003:**
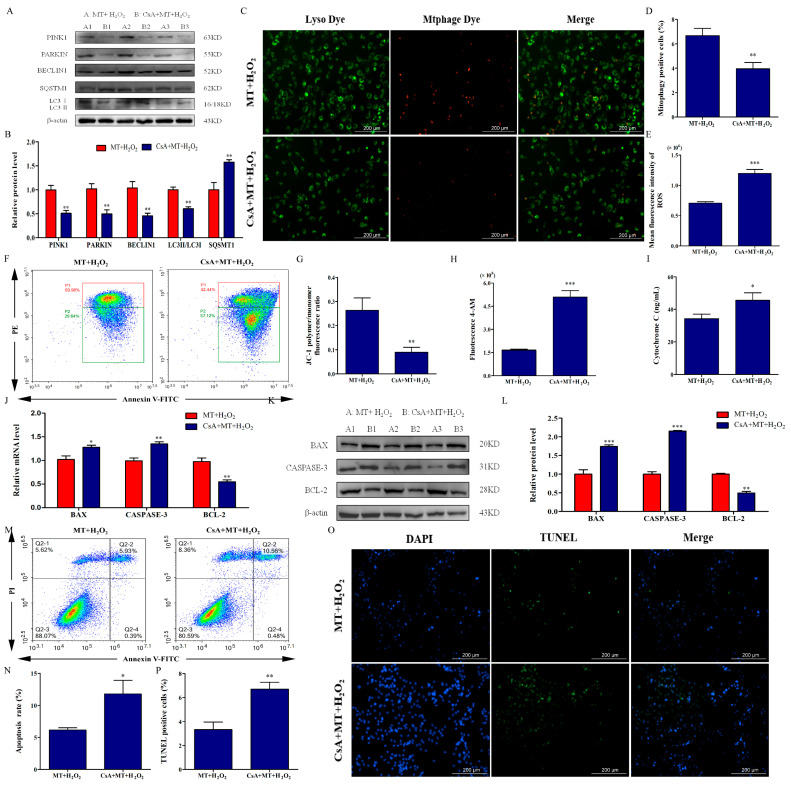
Mitophagy promotion by melatonin attenuates H_2_O_2_-induced mitochondrial injury and apoptosis in bovine ovarian granulosa cells. (**A**,**B**) The relative protein levels of PINK1, PARKIN, BECLIN1, SQSMT1, LC3I, and LC3II were detected by Western blot (*n* = 3). β-actin was used to normalize the density values of the bands. (**C**,**D**) The percentage of mitophagy-positive cells was detected by double staining of lyso dye (green) and mtphage dye (red). (**E**) Mean fluorescence intensity was measured to detect ROS level by flow cytometry assay (*n* = 3). (**F**,**G**) ΔΨm was detected using the dye JC-1 (*n* = 3). The conversion of polymers (red) and monomers (green), and the fluorescence ratio, were measured to analyze ΔΨm. The color from red to blue indicates a decrease in cell density. (**H**) The concentrations of intracellular Ca^2+^ were detected by flow cytometry assay with Fluo-4 AM (*n* = 3). (**I**) The release of Cytochrome C was measured by ELISA (*n* = 3). (**J**) The relative mRNA levels of BAX, CASPASE-3, and BCL-2 were detected by qRT-PCR. (**K**,**L**) The relative protein levels of BAX, CASPASE-3, and BCL-2 were detected by Western blot (*n* = 3). β-actin was used to normalize the density values of the bands. (**M**,**N**) The apoptosis level of granulosa cells was measured by flow cytometry (*n* = 3). Apoptosis rate was expressed as the sum of the values of Q2-2 (late apoptotic cells) and Q2-4 (early apoptotic cells) in the four quadrants. The conversion of polymers and monomers, and the fluorescence ratio, were measured to analyze ΔΨm. (**O**,**P**) The ratio of TUNEL-positive cells was detected by double staining of TUNEL (green) and DAPI (blue). The values are shown as mean ± SD. Differences between groups were determined by Student’s *t*-test. * *p* < 0.05, ** *p* < 0.01, or *** *p* < 0.001 vs. MT+H_2_O_2_ group.

**Figure 4 ijms-24-12854-f004:**
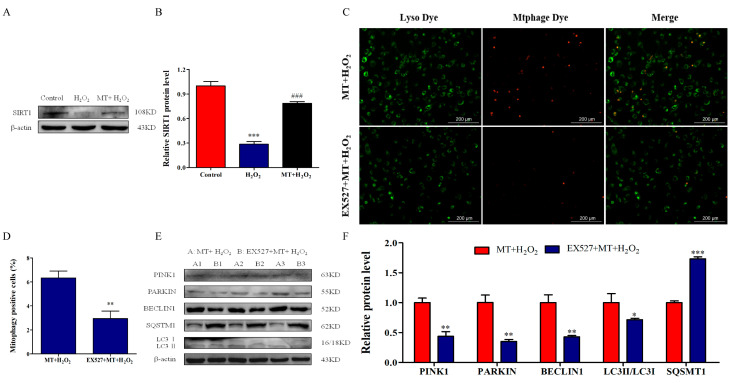
SIRT1 activation by melatonin promotes mitophagy in bovine ovarian granulosa cells. (**A**,**B**) The relative protein level of SIRT1 was detected by Western blot (*n* = 3). *** *p* < 0.001 vs. Control group. ### *p* < 0.001 vs. H_2_O_2_ group. (**C**,**D**) The percentage of mitophagy-positive cells was detected by double staining of lyso dye (green) and mtphage dye (red). (**E**,**F**) The relative protein levels of PINK1, PARKIN, BECLIN1, SQSMT1, LC3I, and LC3II were detected by Western blot (*n* = 3). β-actin was used to normalize the density values of the bands. The values are shown as mean ± SD. * *p* < 0.05, ** *p* < 0.01, or *** *p* < 0.001 vs. MT+H_2_O_2_ group.

**Figure 5 ijms-24-12854-f005:**
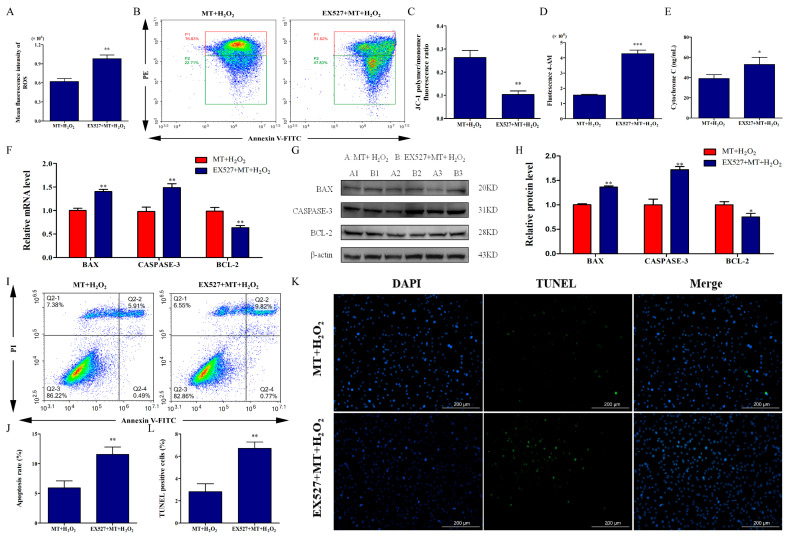
SIRT1 activation by melatonin attenuates H_2_O_2_-induced mitochondrial injury and apoptosis in bovine ovarian granulosa cells. (**A**) Mean fluorescence intensity was measured to detect ROS level by flow cytometry (*n* = 3). (**B**,**C**) ΔΨm was detected using the dye JC-1. The conversion of polymers (red) and monomers (green), and the fluorescence ratio, were measured to analyze ΔΨm. The color from red to blue indicates a decrease in cell density. (**D**) The concentrations of intracellular Ca^2+^ were detected by flow cytometry assay with Fluo-4 AM (*n* = 3). (**E**) The release of Cyt-c was measured by ELISA (*n* = 3). (**F**) The relative mRNA levels of BAX, CASPASE-3, and BCL-2 were detected by qRT-PCR (*n* = 3). (**G**,**H**) The relative protein levels of BAX, CASPASE-3, and BCL-2 were detected by Western blot (*n* = 3). β-actin was used to normalize the density values of the bands. (**I**,**J**) The apoptosis level of granulosa cells was measured by flow cytometry assay (*n* = 3). Apoptosis rate was expressed as the sum of the values of Q2-2 (late apoptotic cells) and Q2-4 (early apoptotic cells) in the four quadrants. The color from red to blue indicates a decrease in cell density. (**K**,**L**) The ratio of TUNEL-positive cells was detected by double staining of TUNEL (green) and DAPI (blue). The values are shown as mean ± SD. * *p <* 0.05, ** *p <* 0.01, or *** *p <* 0.001 vs. MT+H_2_O_2_ group.

**Figure 6 ijms-24-12854-f006:**
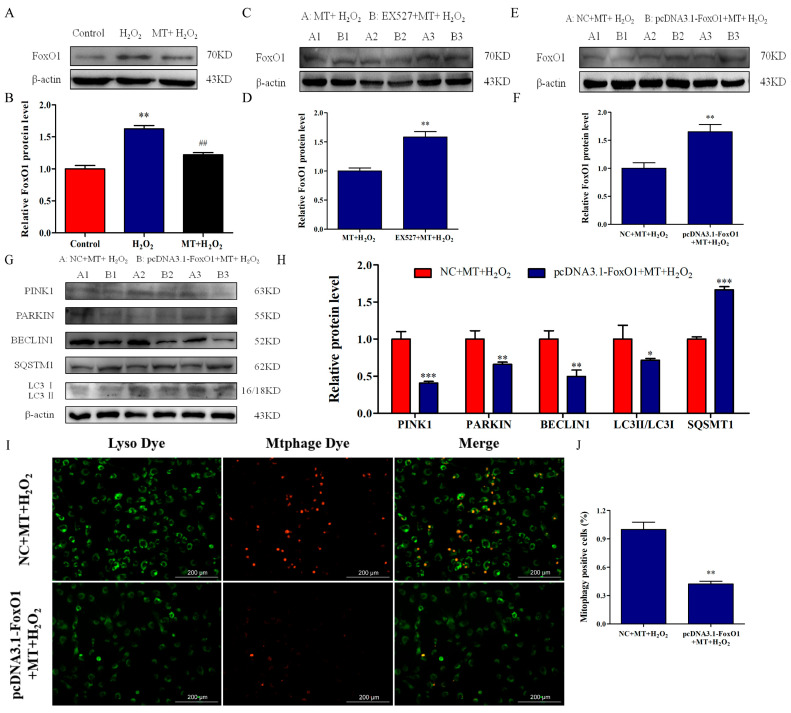
SIRT1/FoxO1 activation by melatonin promotes mitophagy in bovine ovarian granulosa cells. (**A**,**B**) The relative protein level of FoxO1 was detected by Western blot (*n* = 3). ** *p* < 0.01 vs. Control group. ## *p* < 0.01 vs. H_2_O_2_ group.(**C**,**D**) The relative protein level of FoxO1 was detected after treatment with EX527 (*n* = 3). (**E**,**F**) The relative protein level of FoxO1 was detected after overexpression of FoxO1 (*n* = 3). (**G**,**H**) The relative protein levels of PINK1, PARKIN, BECLIN1, SQSMT1, LC3I, and LC3II were detected by Western blot (*n* = 3). (**I**,**J**) The percentage of mitophagy-positive cells was detected by double staining of lyso dye (green) and mtphage dye (red). The values are shown as mean ± SD. * *p <* 0.05, ** *p <* 0.01, or *** *p <* 0.001 vs. MT+H_2_O_2_ group.

**Figure 7 ijms-24-12854-f007:**
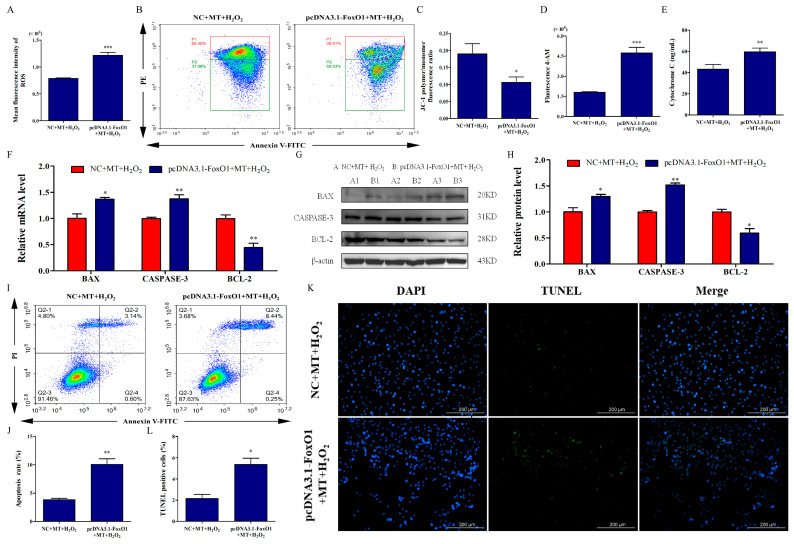
SIRT1/FoxO1 activation by melatonin attenuates H_2_O_2_-induced mitochondrial injury and apoptosis in bovine ovarian granulosa cells. (**A**) Mean fluorescence intensity was measured to detect ROS level by flow cytometry assay *(n* = 3). (**B**,**C**) ΔΨm was detected using the dye JC-1 (*n* = 3). The conversion of polymers (red) and monomers (green), and the fluorescence ratio, were measured to analyze ΔΨm. The color from red to blue indicates a decrease in cell density. (**D**) The concentrations of intracellular Ca^2+^ were detected by flow cytometry assay with Fluo-4 AM. (**E**) The release of Cyt-c was measured by ELISA. (**F**) The relative mRNA levels of BAX, CASPASE-3, and BCL-2 were detected by qRT-PCR (*n* = 3). (**G**,**H**) The relative protein levels of BAX, CASPASE-3, and BCL-2 were detected by Western blot (*n* = 3). (**I**,**J**) The apoptosis level of granulosa cells was measured by flow cytometry (*n* = 3). Apoptosis rate was expressed as the sum of the values of Q2-2 (late apoptotic cells) and Q2-4 (early apoptotic cells) in the four quadrants. The color from red to blue indicates a decrease in cell density. (**K**,**L**) The ratio of TUNEL-positive cells was detected by double staining of TUNEL (green) and DAPI (blue). The values are shown as mean ± SD. * *p <* 0.05, ** *p <* 0.01, or *** *p <* 0.001 vs. MT+H_2_O_2_ group.

**Table 1 ijms-24-12854-t001:** Primer information for qRT-PCR.

Gene	Genebank No.	Primer Sequence (5′-3′)	Size (bp)
BCL-2	NM_001166486.1	F: TGGATGACCGAGTACCTGAACCG R: TGCCTTCAGAGACAGCCAGGAG	132
BAX	NM_173894.1	F: GGCTGGACATTGGACTTCCTTCG R: ATGGTGAGCGAGGCGGTGAG	149
CASPASE-3	NM_001077840.1	F: GACAGACAGTGGTGCTGAGG R: AGAAACATCACGCATCAA	151
β-actin	NM_173979.3	F: TTGATCTTCATTGTGCTGGGTG R: CTTCCTGGGCATGGAATCCT	189

**Table 2 ijms-24-12854-t002:** Details of primary and secondary antibodies used for Western blot.

Antibodies	Cat No.	Source	Dilution
BCL-2	A11025	ABclonal, Wuhan, China	1:1000
BAX	A12009	ABclonal, Wuhan, China	1:1000
PINK1	orb36596	Biorbyt, Cambridge, UK	1:500
PARKIN	orb36592	Biorbyt, Cambridge, UK	1:500
BECLIN1	ab207612	Abcam, Cambridge, UK	1:1000
LC3	AL221	Beyotime, Shanghai, China	1:1000
SQSTM1	ab207305	Abcam, Cambridge, UK	1:1000
SIRT1	ab110304	Abcam, Cambridge, UK	1:1000
FoxO1	ab179450	Abcam, Cambridge, UK	1:1000
β-actin	60008-1-lg	ProteinTech, Chicago, IL, USA	1:6000
Goat Anti-Rabbit IgG(H+L)	SA00001-2	ProteinTech, Chicago, IL, USA	1:10,000
Goat Anti-Mouse IgG(H+L)	SA00001-1	ProteinTech, Chicago, IL, USA	1:10,000

## Data Availability

The data presented in this study are available on request from the corresponding author.

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
