# Peer review of "Melatonin Attenuates Oxidative Stress-Induced Apoptosis of Bovine Ovarian Granulosa Cells by Promoting Mitophagy via SIRT1/FoxO1 Signaling Pathway"

_ijms, 2023, doi:10.3390/ijms241612854_

Round 1

Reviewer 1 Report (Previous Reviewer 3)

The work of Gaoqing Xu et. al. was to investigate the effects of melatonin on mitophagy and apoptosis of bovine ovarian granulosa cells under oxidative stress,and to clarify this mechanism. I believe that the research topic is interesting and important, but the theoretical introduction could explain the ovulation processes and the impact of stress on ovulation disorders in more detail. The results were presented very clearly. The quality of the graphs and figures is sufficient. However, I think that the description of the methodology could be more detailed. I would like to ask for a justification for the choice of the tested concentrations and information on how many repetitions the experiment was performed. A description of the principle of the method would also help to understand the selected studies. In the discussion, please expand on the topic of the antioxidant effect of melatonin and describe the mitophagy process in detail - what can induce it, how it proceeds, in what processes it occurs. I congratulate the authors on undertaking an interesting research topic. After minor revision, the work can be published

I am not qualified to assess the quality of English in this paper

Author Response

Reviewer 2 Report (New Reviewer)

Line numbering is lacking so it is quite difficult to indicate criticisms

Abstract:

granulosa instead granule

suggest instead of suggested

Introduction

atretic instead of atresis

largest group: please change this words

ROS can promote oxidative stress: please revise the sentence

ROS can be also signalling molecules: plese add info

Please add and discuss the findings from these papers:

Melatonin modulates swine luteal and adipose stromal cell functions.

Dodi A, Bussolati S, Grolli S, Grasselli F, Di Lecce R, Basini G.

Reprod Fertil Dev. 2021 Feb;33(3):198-208. doi: 10.1071/RD20312.

Melatonin potentially acts directly on swine ovary by modulating granulosa cell function and angiogenesis.

Basini G, Bussolati S, Ciccimarra R, Grasselli F.

Reprod Fertil Dev. 2017 Nov;29(12):2305-2312. doi: 10.1071/RD16513.

Please explain che choice of the bovine model

Methods:

FCS luteinizes granulosa cells. Please explain the choice of this culture method

Add reference for the examined concentrations of melatonin and H2O2 mainly in relation to physiological levels.

English language is fine

Author Response

This manuscript is a resubmission of an earlier submission. The following is a list of the peer review reports and author responses from that submission.

Round 1

Reviewer 1 Report

The manuscript entitled "Melatonin attenuates oxidative stress-induced apoptosis of bovine ovarian granulosa cells by promoting mitophagy via SIRT1/FoxO1 signaling pathway" in which the authors investigated the effects of melatonin on mitophagy and apoptosis of bovine ovarian granulosa cells under oxidative stress. Their findings demonstrated that melatonin could promote mitophagy to attenuate oxidative stress-induced apoptosis and mitochondrial dysfunction of bovine ovarian granulosa cells via SIRT1/FoxO1 signaling pathway.

The work is understandable and the topic is important. The scientific narrative is well structured and flows naturally from one idea to the next. The results are interesting.

However, this paper suffers from some shortcomings that if modified would make the manuscript very suitable for publication in International Journal of Molecular Sciences.

Major concern:

1-      Please add the number of sample in each figure legend.

2-      The authors write “Differences between groups were determined by Student's t-test and one-way analysis of variance (ANOVA)”. Please add if they used Student's t-test or ANOVA in each figure legend.

3-      Did the author normalize their PCR data to β-actin? Please mention the way of calculation if by using the 2-ΔΔCt method for example in methods section.

4-      Please write the way of detection of antibodies in western blot, method of analysis, and the calculation of the intensity of the bands of target proteins in methods section.

5-      If the target protein levels in western blot were normalized to β-actin or tubulin or any other housekeeping protein, please write that in figure legend of western blot data.

6-      In figures of western blot data, did the authors use the same membrane to check the different proteins in each experiment? please clarify?

7-      In figures 3A, 4B, PINK1, PARKIN, BECLIN1 protein levels were very faint. Are these proteins measured in the mitochondria of cells? If the way of extraction is similar as mentioned in western blot by RIPA buffer or by another method? Please clarify?

Minor concern:

1.      I think if you combine the whole blot of each western blot in one supplementary pdf file, it will be less confusing. Adding the figure number, molecular weight, name of proteins, and names of different groups should be done.

2.      I think if you make a graphical abstract or figure of conclusion of your findings, it will enrich your manuscript.

Reviewer 2 Report

The manuscript by Qiu et al. examined the role of mitophagy in the protection of melatonin on bovine ovarian granulosa cells and the mechanism of melatonin regulating mitophagy. They found that melatonin attenuates oxidative stress-induced mitochondrial dysfunction and apoptosis, and promotes PINK1/PARKIN-mediated mitophagy of bovine ovarian granulosa cells. The major conclusions of this research are justified by the results. The methodology seems to be correct in most experiments and the results of this work may be worth publishing. However, the study requires improvement in some aspects. Please consider the following points:

Minor revision:

There are so many grammatically misspelled. ‎

 For example

In line 87, inhibited should be replaced by decreased;

In line 91, cells should be replaced by cell;

In line 97, apoptosis should be replaced by apoptotic;

In line 107, afainst was spell mistake.

Reviewer 3 Report

The work of Gaoqing Xu et a. presents a very interesting topic of protection of ovarian granulosa cells against the harmful effects of oxidative stress. Scientists are assessing whether the use of melatonin may have a protective effect on reproductive cells subjected to oxidative stress. I believe that the research topic is interesting, the authors used the correct methodology and obtained interesting results, although I have a few comments:

1. I believe that the presentation of the results should be considered. Grouping several different graphics into one chart adversely affects the readability of the results. The resolution of the presented photos is insufficient. Especially images for WesternBlot and TUNEl should be presented more clearly.

2. I would like to ask why - according to the authors - the viability of the tested cells decreased when using H2O2 and 1 or 10 μM melatonin?

3. On the basis of what data did the authors choose the duration of the experiment (24h incubation with melatonin, addition of H2O2 for 4h)?

4. I believe that each determination would be worth repeating at least 6 times

5. I would like to ask for a broader description of the principles of cell culture (were the cells adherent? Under what conditions were they grown? Were antibiotics added to the culture? Were the cells passaged and how?)

6. How many cells were analyzed in the cytometer?

7. In the discussion, it is worth comparing your results to the results of research by other scientists.

Please also check for minor typos.

Thanking you for the opportunity to get acquainted with the work, I wish the authors further scientific success.
